# LLaMaFlex: Many-in-one LLMs via Generalized Pruning and Weight Sharing

**Ruisi Cai**[1,2], **Saurav Muralidharan**[1]\*, **Hongxu Yin**[1],
**Zhangyang Wang**[2], **Jan Kautz**[1], **Pavlo Molchanov**[1]
[1] NVIDIA    [2] The University of Texas at Austin

## Abstract

Large Language Model (LLM) providers typically train a family of models, each of a different size targeting a specific deployment scenario. Models in the family are all trained from scratch, making the process extremely resource intensive. Recent work has successfully reduced the cost of training model families through a combination of structured pruning and knowledge distillation; here, only the largest model in the family is trained from scratch, and smaller models are obtained via pruning. We observe that while effective, this strategy must still perform pruning and distillation with hundreds of billions of training tokens for every new model, keeping overall training costs high. In this work, we introduce a novel nested weight-shared architecture named LLaMaFlex that can be pruned across both width and depth dimensions *in a zero-shot* manner to instantly yield a large number of highly accurate compressed models. LLaMaFlex starts from a pre-trained model, and only requires a single continued training phase consisting of $\sim 60B$ tokens, which trains the elastic network and an end-to-end Gumbel Softmax-based router; this router is able to interpolate smoothly across model sizes, enabling the "train once, deploy many" paradigm. We train LLaMaFlex on Llama 3.1 8B and use it to zero-shot generate a family of compressed models that achieves accuracy on par with or better than state-of-the-art pruned, elastic/flexible, and trained-from-scratch models.

## 1 Introduction

Large Language Models (LLMs) have demonstrated remarkable effectiveness across a diverse range of tasks (Hendrycks et al., 2020; Chiang et al., 2024; Zheng et al., 2024). However, the generality and robustness of LLMs are largely attributed to their vast scale, with parameter counts ranging from one billion to several hundred billion (Touvron et al., 2023; Zhang et al., 2022). This substantial model size, in turn, makes them extremely resource-intensive to train and deploy. This problem is further exacerbated in model deployment scenarios characterized by varying compute and memory requirements, requiring model providers to train an entire LLM "family", containing different-sized models for each scenario; for instance, the Llama 3.1 model family includes three different variants with 8, 70 and 405 billion parameters, each trained from scratch on 15 trillion tokens (Dubey et al., 2024).

Recent work has attempted to reduce the training cost of LLM families via a combination of structured pruning and knowledge distillation (Muralidharan et al., 2024); in this setting, only the largest model in the family is trained from scratch, with smaller models obtained via compression. Specifically, the original pretrained model is pruned along both depth (layers) and width (attention heads, hidden dimension, MLP intermediate size) dimensions to target a particular parameter budget; this pruned network is then distilled with

---

\*Correspondence to: `sauravm@nvidia.com`

the original model acting as teacher. While this approach succeeds in reducing both the compute and number of training tokens required (w.r.t. training each model from scratch), it suffers from a major drawback: each compressed model targeting a particular parameter or latency budget must be distilled separately; this keeps the training cost for LLM families very high. Instead, we ask the following question: *is it possible to create a flexible network architecture which can yield different pruned sub-networks (depending on budget) without the need for any additional fine-tuning or distillation - in a single shot?* Our answer to this question is a new elastic LLM architecture and post-training optimization framework named LLAMAFLEX.

LLAMAFLEX utilizes a nested weight-shared network architecture which can be sliced along both depth (layers) and width (attention heads, hidden dimension, MLP intermediate size) axes, enabling on-demand, zero-shot generation of accurate pruned models at inference time. An end-to-end trained router is learned to dynamically decide the optimal network architecture given a parameter budget. We start from a pretrained model, and perform a brief elastic pretraining and router training (60.4B training tokens in this work). LLAMAFLEX is able to generate a large number of pruned networks *without the need for any additional fine-tuning or distillation*. As shown in Figure 1, the zero-shot pruned models obtained with LLAMAFLEX perform on par with or better than models obtained via structured pruning and distillation; we also notice that LLAMAFLEX models outperform similarly-sized models trained from scratch. Figure 2 provides an overview of how LLAMAFLEX models are created, and compares our approach to (1) training from scratch, and (2) pruning + distillation.

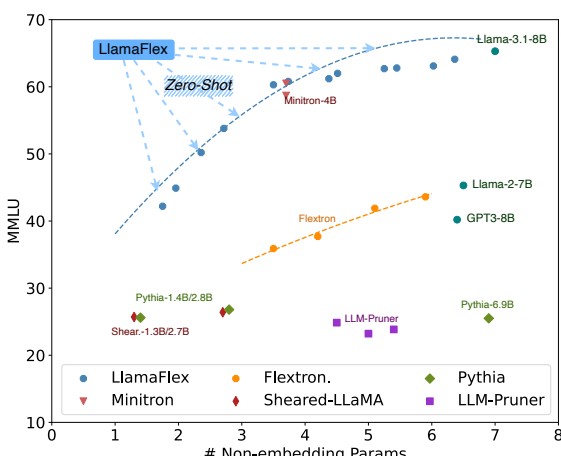

Figure 1: MMLU vs. model size for LLAMAFLEX vs. other similar frameworks/models. LLAMAFLEX enables zero-shot generation of a Pareto frontier of models that beats the accuracy of current state-of-the-art compressed and trained-from-scratch models.

Elastic nested transformer architectures are not new and have been explored in the past. Specifically, Matformer (Kudugunta et al., 2023) and Flextron (Cai et al., 2024) use a nested weight-shared architecture that is very similar to the one used in LLAMAFLEX; Flextron also includes an input-adaptive router that can select some width axes (MLP intermediate size and number of attention heads) based on a target latency. We note that LLAMAFLEX differs in several key ways: (1) to the best of our knowledge, no existing elastic training

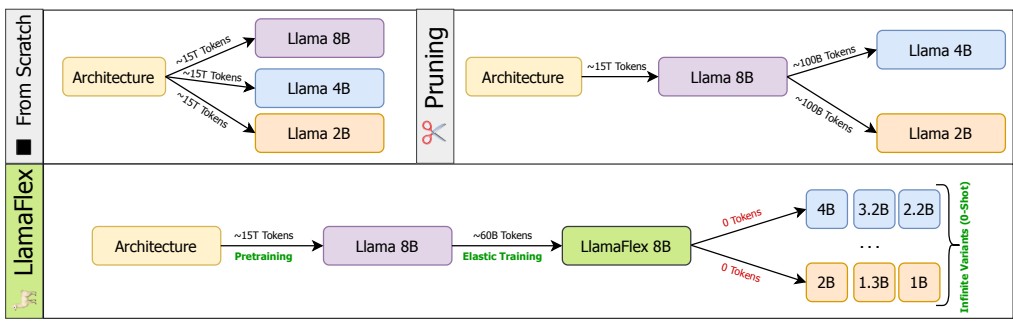

Figure 2: High-level overview of LLAMAFLEX. In contrast to training from scratch (top left part of the figure) and pruning (top right), LLAMAFLEX supports the generation of a potentially infinite number of pruned models without additional fine-tuning/distillation (zero-shot) after a brief elastic pretraining phase.

framework supports the full suite of width and depth dimensions other than LLAMAFLEX, (2) pure nesting introduces a hard constraint on the resulting models which may potentially make them less expressive; we counter this by introducing additional parameters for each nesting level through a technique we term *policy-aware modulation*, and (3) both Matformer and Flextron produce heterogenous architectures (each layer has potentially different architectural configurations) which are difficult to deploy in common frameworks such as TensorRT-LLM (NVIDIA, 2023) and llama.cpp; LLAMAFLEX always produces uniform architectures which have optimized implementations in such common LLM deployment frameworks.

This paper makes the following key contributions:

1. Introduces a novel elastic LLM architecture named LLAMAFLEX which can be resized along both depth and width axes to achieve superior accuracy (compared to other pruning approaches, nested architectures, and training from scratch) for a given deployment scenario with no fine-tuning required. Further, LLAMAFLEX produces pruned architectures that are uniform, making it easy to deploy using existing LLM frameworks such as TensorRT-LLM.

2. Presents a Gumbel Softmax-based end-to-end learnable router that is able to interpolate smoothly from 25% to 100% of the original model's size in a zero-shot manner. This enables the *"train once, generate many"* paradigm.

3. Increases the generality of nested architectures by introducing a policy-aware modulation technique inspired from the diffusion model literature.

4. Produces a state-of-the-art family of compressed models from Llama 3.1 8B, each of which can be obtained zero-shot after an initial elastic pretraining and router training phase consisting of $\sim 60B$ tokens. LLAMAFLEX achieves better accuracy than state-of-the-art compression techniques such as Minitron Muralidharan et al. (2024), nested weight-shared architectures such as Flextron Cai et al. (2024) and models trained from scratch.

## 2 METHOD

In the following sections, we will cover the design of LLAMAFLEX and go over the process of converting a pretrained LLM to an elastic form. We provide a high-level overview of LLAMAFLEX's design in Figure 3.

### 2.1 ELASTIC FRAMEWORK: BACKGROUND AND NOTATION

Large Language Models (LLMs) process diverse inputs and tasks by passing tokens through stacked transformer blocks, whose shapes are characterized by some key *architectural variables*: the number of transformer blocks ($N$), the hidden dimension size ($H$), the intermediate dimension of the MLP layer ($D$), and the number of attention heads ($N_A$). Collectively, these parameters, denoted as $(D, N_A, H, N)$, describe the "shape" of the model. Specifically, an LLM with shape $(D, N_A, H, N)$ models language as:

$$\text{LLM}_{(D,N_A,H,N)}(\boldsymbol{x}) = \boldsymbol{x}_N \quad \text{where}$$

$$\boldsymbol{x}_{i+1} = \left(\text{MHA}^i_{(H,N_A)} \circ \text{MLP}^i_{(H,D)}\right)(\boldsymbol{x}_i) + \boldsymbol{x}_i, \quad i = 1, \ldots, N-1 \tag{1}$$

LLAMAFLEX allows the original model to operate as diverse model variants, each with a unique shape and achieving a different performance-efficiency trade-off. Users are able to choose the variant with a suitable size during inference, based on their resource and performance constraints. Specifically, a model variant $j$ with shape $(D^j, N_A^j, H^j, N^j)$ can be formalized as follows:

$$\text{Elastic-LLM}_{(D^j,N_A^j,H^j,N^j)}(\boldsymbol{x}; \boldsymbol{\lambda}^j) = \boldsymbol{x}_N \quad \text{where}$$

$$\boldsymbol{x}_{i+1} = \lambda_i^j \left(\text{MHA}^i_{(H^j,N_A^j)} \circ \text{MLP}^i_{(H^j,D^j)}\right)(\boldsymbol{x}_i) + \boldsymbol{x}_i, \quad i = 1, \ldots, N-1, \tag{2}$$

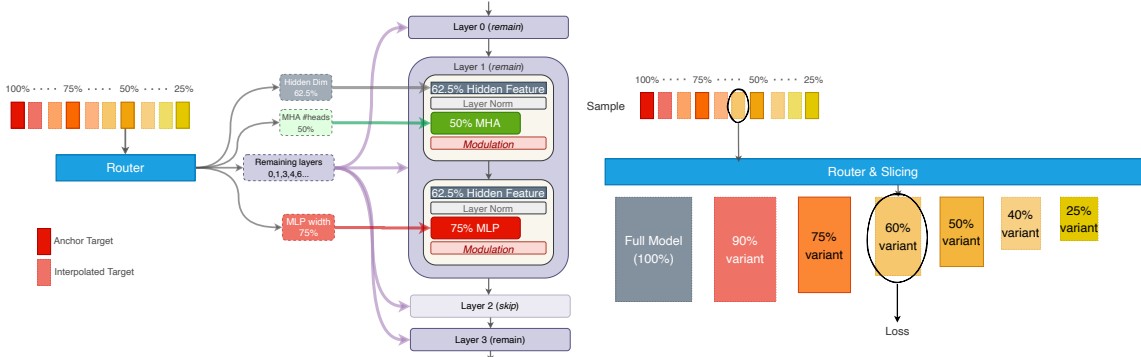

Figure 3: Overview of the LLAMAFLEX Framework. A router is introduced to determine the sub-network configuration (i.e., the size of the MLP, MHA, hidden dimensions, and the number of layers) that satisfies a given budget target (i.e., the percentage of remaining parameters) while optimizing performance. Although the router is trained on discrete "anchor" budget targets (e.g., 50%, 75%, as shown), it generalizes well to unseen budget targets (e.g., 62.5%).

here, $\lambda_i^j$ is a binary scaler, controlling whether layer $i$ in submodel $j$ is skipped ($\lambda_i^j = 0$) or not. $\lambda_i^j$ is the $i$-th item of $\boldsymbol{\lambda}^j$; $\sum_i \lambda_i^j = N^j$. The elastic MHA and MLP layers are defined as follows:

$$\text{MLP}^i_{(H^j, D^j)}(\boldsymbol{x}_i) = \sigma\left(\boldsymbol{x}_i \cdot \left(\mathbf{I}_{D^j} \boldsymbol{W}^{(1)} \mathbf{I}_{H^j}\right)^T\right) \cdot \left(\mathbf{I}_{D^j} \boldsymbol{W}^{(2)} \mathbf{I}_{H^j}\right),$$

$$\text{MHA}^i_{(H^j, N_A^j)}(\boldsymbol{x}_i) = \text{Concat}(\text{head}_1, ... \text{head}_{N_A^j}) \cdot \left(\mathbf{I}_{N_A^j C} \boldsymbol{W}^O\right), \tag{3}$$

$$\text{head}_k = \text{Attn}(\boldsymbol{x}_i \mathbf{I}_{H^j} \boldsymbol{W}_k^Q, \boldsymbol{x}_i \mathbf{I}_{H^j} \boldsymbol{W}_k^K, \boldsymbol{x}_i \mathbf{I}_{H^j} \boldsymbol{W}_k^V),$$

where, $\mathbf{I}_{D^j} \in \mathbb{R}^{D \times D}$, $\mathbf{I}_{H^j} \in \mathbb{R}^{H \times H}$, and $\mathbf{I}_{N_A^j C} \in \mathbb{R}^{H \times H}$ are diagonal matrices with the first $D^j$, $H^j$, and $N_A^j C$ diagonal elements equal to 1, and the rest 0, respectively. $C$ denotes the size of a single head. These diagonal matrices ensure that the $j^{\text{th}}$ sub-model only utilizes the first $H^j$ hidden features, the first $D^j$ MLP intermediate neurons, and the first $N_A^j$ attention heads. We constrain $D^j < D$, $H^j < H$, $N_A^j < N_A$. $\boldsymbol{W}^{(1)}$ and $\boldsymbol{W}^{(2)}$ are the associated two weight matrices in MLP layers, with $\boldsymbol{W}^{(1)}, \boldsymbol{W}^{(2)} \in \mathbb{R}^{D \times H}$; $\sigma(\cdot)$ refers to the non-linear activation function. $\boldsymbol{W}^{Q,i}, \boldsymbol{W}^{K,i}, \boldsymbol{W}^{V,i} \in \mathbb{R}^{H \times C}$ and $\boldsymbol{W}^O \in \mathbb{R}^{N_A C \times H}$. For implementation, the diagonal matrix $\mathbf{I}$ can be replaced with a slicing operator. Compared to existing methods (Kudugunta et al., 2023; Kavehzadeh et al., 2023; Cai et al., 2024), LLAMAFLEX is the first generalized elastic framework that supports learnable layer skipping and flexible hidden dimension.

## 2.2 GENERALIZABLE ROUTER DESIGN

**Problem Formulation**  The router is tasked with identifying the optimal configuration of the sub-model $(D^j, N_A^j, H^j, N^j)$ achieving the best performance while adhering to budget constraints $b_j$, defined as the percentage of remaining parameters relative to the original model. This process can be formalized as a combinatorial optimization problem:

$$\min_{(D^j, N_A^j, H^j, N^j, \boldsymbol{\lambda}^j)} \mathbb{E}_{x \sim p(x)} \left[ \mathcal{L}_{(D^j, N_A^j, H^j, N^j)}(\boldsymbol{x}; \boldsymbol{\lambda}^j) \right],$$

$$\text{s.t.} \quad \mathcal{P}(D^j, N_A^j, H^j, N^j, \boldsymbol{\lambda}^j) \leq b_j \cdot \mathcal{P}(D, N_A, H, N, \mathbf{1}_N), \tag{4}$$

$$D^j \in \mathcal{D}, \quad N_A^j \in \mathcal{N}_A, \quad H^j \in \mathcal{H}, \quad N^j \in \mathcal{N},$$

where $\mathcal{P}(\cdot)$ denotes the number of parameters given the network dimensions (i.e., $D^j, N_A^j, H^j, N^j$) and the binary vector for layer skipping (i.e., $\boldsymbol{\lambda}^j$). $\mathcal{P}(D, N_A, H, N, \mathbf{1}_N)$ denotes the number of parameters in the original pre-trained model. To satisfy the constraint, we introduce an additional loss term, which is detailed in Sec 2.3. Each of the dimensions are selected from the pre-defined sets $\mathcal{D}, \mathcal{N}_A, \mathcal{H}, \mathcal{N}$, respectively.

**Router Design.** Due to the discrete nature of the variables $D^j, N_A^j, H^j, N^j, \{\lambda_i^j, i \in [0, N]\}$, traditional gradient descent methods are ineffective as they require the continuous differentiability of the objective function with respect to the variables. Direct optimization of these discrete choices would typically involve an exhaustive search or evolutionary algorithms, which can be computationally prohibitive given the large search space. To address this challenge, we utilize the Gumbel-Softmax trick (Jang et al., 2016) as a continuous relaxation of the discrete optimization problem. The Gumbel-Softmax technique allows us to approximate categorical distributions over discrete choices with a differentiable softmax function, enabling the use of gradient-based optimization methods. We first represent each architectural variable as a categorical distribution, and instead of directly selecting values, we model the probability of each choice. For example, for $D^j$, we define a categorical distribution with class probabilities $P(D^j = m_d)$ for $m_d \in \mathcal{D}$, where the probabilities satisfy the constraint $\sum_{m_d} P(D^j = m_d) = 1$. During stochastic sampling, the algorithm assigns higher probabilities to values of $m_d$ that are more likely to produce better performance, reflecting a preference for favorable configurations.

**Router Architecture** We model the categorical distribution of each architectural variable using a lightweight router implemented as a two-layer MLP. As described in Section 2.1, architectural variables include the MLP intermediate dimension $D^j$, hidden dimension $H^j$, number of heads in MHA $N_A$, and the Bernoulli variable $\lambda_i^j$, which controls whether to skip the layer $i$ or not. For each budget $b_j \in \mathcal{B}$, we first encode it as a one-hot vector, $\boldsymbol{h}_j = \text{One-Hot}(b_j)$, where $\boldsymbol{h}_j \in \mathbb{R}^{|\mathcal{B}| \times 1}$. For instance, to model the distribution for $D^j$, we input $\boldsymbol{h}_j$ into the MLP and obtain the un-normalized log-probilities $\log \pi(D^j = m_d) = \text{MLP}_D(\boldsymbol{h}_j)$. Similarly, we compute the logits for the other architectural variables: $\log \pi(H^j = m_h) = \text{MLP}_H(\boldsymbol{h}_j), \log \pi(N_A^j = m_a) = \text{MLP}_{N_A}(\boldsymbol{h}_j)$, and $\log \pi(\lambda_i^j = m_\lambda) = \text{MLP}_{\lambda_i}(\boldsymbol{h}_j)$. This framework allows the router to generalize to unseen budget targets, $b_k \notin \mathcal{B}$, a crucial capability we will discuss in more detail later in this Section.

**Router Optimization** To enable differentiable sampling, we apply the Gumbel-Softmax (Jang et al., 2016) trick to these categorical distributions. For each architectural variable, the categorical distribution is reparameterized as:

$$P(D^j = m_d) = \frac{\exp\left((\kappa \log \pi(D^j = m_d) + g_d)/\tau\right)}{\sum_{m \in \mathcal{D}} \exp\left((\kappa \log \pi(D^j = m) + g)/\tau\right)}, \tag{5}$$

where $g_d$ is a sample from the Gumbel(0,1) distribution, and $\tau$ is a temperature parameter that controls the smoothness of the approximation; $\kappa$ is the scaling factor to balance the relative magnitude of logits and Gumbel noises. As $\tau \to 0$, the distribution approaches a one-hot vector, allowing the router to make discrete choices. With the same formulation, we compute $P(H^j = m_h)$, $P(N_A^j = m_a)$, and $P(\lambda_i^j = m_\lambda)$. The optimization problem now focuses on minimizing the following loss function:

$$\mathcal{L}_R(\boldsymbol{x}; \theta, \theta_R, j) = \mathbb{E}_{(D^j, N_A^j, H^j, N^j, \boldsymbol{\lambda}^j) \sim \mathcal{Q}_{\theta_R}(\cdot|j)}\left[\mathcal{L}_{(D^j, N_A^j, H^j, N^j)}(\boldsymbol{x}; \theta, \boldsymbol{\lambda}^j)\right], \tag{6}$$

where $\theta_R$ represents the parameters of the router and $\theta$ denotes the parameters of the backbone. $\mathcal{Q}_{\theta_R}$ indicates that the architectural choices $(D^j, N_A^j, H^j, N^j, \boldsymbol{\lambda}^j)$ are sampled from the probability distributions modeled by the routers, for budget target $b_j$, which is parameterized by $\theta_R$.

**Router Interpolation** LLAMAFLEX allows the router to generalize to unseen budget targets at zero additional cost. This is a crucial capability that in turn allows LLAMAFLEX to instantly generate pruned models

targeting specific user-defined deployment targets. As previously discussed, when $b_j \in \mathcal{B}$, the embedding vector is obtained through one-hot encoding, $\boldsymbol{h}_j = \text{One-Hot}(b_j)$. For an unseen budget target $b_k \notin \mathcal{B}$, its embedding vector can be derived by linearly interpolate between its nearest neighbors in the budget set. Specifically, Assume that $b_n$ and $b_{n+1}$ are the nearest known budget targets such that $b_n \le b_k \le b_{n+1}$. The embedding vector $\boldsymbol{h}_k$ for the unseen budget target $b_k$ can be interpolated linearly as follows:

$$\boldsymbol{h}_k = \frac{b_{n+1} - b_k}{b_{n+1} - b_n} \cdot \boldsymbol{h}_n + \frac{b_k - b_n}{b_{n+1} - b_n} \cdot \boldsymbol{h}_{n+1} \tag{7}$$

This linear interpolation ensures a smooth transition between the embedding vectors of the known budgets, enabling the router to generalize effectively to the unseen budget target $b_k$. With this interpolation capability, LLAMAFLEX can produce a spectrum of model variants that adapt to continuously changing budget targets.

### 2.3 MODEL PREPARATION AND ROUTER TRAINING

**Model Preparation**  Given the pre-trained model, we first prepare it following the approach in prior work (Cai et al., 2024; Muralidharan et al., 2024). Specifically, using a small set of data samples, we compute the importance of each MLP neuron, attention head, and hidden feature based on the accumulated magnitude of activations. Once the importance is determined, we reorder the corresponding weight matrices such that neurons, heads, and hidden features are arranged in decreasing order of importance for each layer. Sub-networks can then be constructed by simply selecting the first several neurons or heads in each layer, thereby preserving the essential knowledge encoded in the most important channels.

**Router Training**  After the model preparation step, we conduct end-to-end training to jointly optimize the backbone parameters $\theta$ and router parameters $\theta_R$, using the loss function below:

$$\mathcal{L} = \sum_j \left( \mathcal{L}_R(\boldsymbol{x}_j; \theta, \theta_R, j) + \mathcal{L}_B(\theta_R, j) \right) \tag{8}$$

Here, we note that each sub-network $j$ uses a different data batch $\boldsymbol{x}_j$. Since the sub-networks are created by slicing and therefore share weights, the gradient propagated by one sub-network can implicitly affect the others. This allows all sub-networks to benefit from the information across different data batches, even though they are directly trained on separate inputs. Unlike traditional pruning (Xia et al., 2023; Ma et al., 2023; Muralidharan et al., 2024), which generate pruned variants separately, weight sharing helps training efficiency by allowing sub-networks to collectively leverage knowledge from various inputs. In addition to the router loss $\mathcal{L}_R(\cdot)$, we introduce an additional loss term $\mathcal{L}_B(\cdot)$ to ensure that the sampled sub-network adheres to the parameter budget constraint defined in Equ. 4. Here, $\mathcal{P}(\cdot)$ represents the number of parameters based on the network dimensions (i.e., $D^j, N_A^j, H^j, N^j$) and the binary vector for layer skipping (i.e., $\boldsymbol{\lambda}^j$). Meanwhile, $\mathcal{P}_{\text{full}}$ refers to the total number of parameters in the original pre-trained model.

$$\mathcal{L}_B(\theta_R, j) = \max \left( \mathbb{E}_{(D^j, N_A^j, H^j, N^j, \boldsymbol{\lambda}^j) \sim \mathcal{Q}_{\theta_R}(\cdot|j)} \mathcal{P}(D^j, N_A^j, H^j, N^j, \boldsymbol{\lambda}^j) - b_j \cdot \mathcal{P}_{\text{full}}, 0 \right),$$
$$\text{where} \quad \mathcal{P}_{\text{full}} = \mathcal{P}(D, N_A, H, N, \mathbf{1}_N), 0). \tag{9}$$

Following prior approaches (Fang et al., 2024), we apply an exponential decay to the value of temperature, while linearly increasing the scaling factor of router outputs. We provide more details on hyper-parameter choice in Sec 3.1. In the initial phase, when the temperature is high and the scaling factor is small, significant randomness is introduced, allowing for optimization of the router. As training progresses, the router transitions to making deterministic decisions.

### 2.4 POLICY-AWARE MODULATION

Due to its simplicity, a nested structure has been widely adopted in most existing elastic LLM work (Kusupati et al., 2022; Kudugunta et al., 2023; Kavehzadeh et al., 2023). In this approach, each sub-network is

generated by simply slicing the weight matrices. However, this constraint can potentially limit the representational capacity of elastic networks, as the same set of weights must accommodate inputs across all possible sub-networks.

To address this limitation, we propose the use of *policy-aware modulation*, a technique inspired by methods in the literature on diffusion models (Ho et al., 2020; Peebles & Xie, 2023). In these models, the time step $t$ is encoded in the Transformer layers through learnable normalization, allowing the same parameters to process inputs with varying noise scales. We adopt a similar approach in LLAMAFLEX by introducing lightweight non-

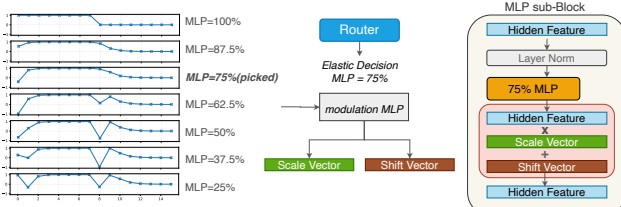

Figure 4: The illustration on Modulation.

linear modulation heads, which are applied after the elastic components (i.e., elastic MLP/elastic MHA, as detailed in Equ. 3). These heads modulate the outputs of elastic operations based on the elastic choice. Taking elastic MLP as an example, if the current sub-network chooses to use $e_k$ hidden neurons, we condition the output of the elastic MLP on $e_k$. Concretely, we use a sinusoidal embedding of $e_k$ followed by a learnable MLP layer to generate modulation vectors for scaling and shifting. These modulation vectors then transform the output of the elastic MLP $\boldsymbol{y}$ as follows:

$$\hat{\boldsymbol{y}} = \boldsymbol{y} \cdot \mathrm{MLP}_{\mathrm{scale}}(\mathrm{Emb}(e_k)) + \mathrm{MLP}_{\mathrm{shift}}(\mathrm{Emb}(e_k)) \tag{10}$$

## 3 RESULTS

### 3.1 EXPERIMENTAL DETAILS

**Pre-trained Model and Dataset** We validate our method on Llama 3.1 8B (Dubey et al., 2024), a prominent open-source large language model trained on 15 trillion tokens. The model comprises $N = 32$ Transformer blocks, each containing a Multi-head Attention (MHA) layer with 8 attention groups and $N_A = 32$ query heads, along with a Multilayer Perceptron (MLP) with an intermediate dimension of $D = 14436$. The hidden feature dimension is $H = 4096$. The model has a total of 6.98 billion non-embedding parameters, with 5.64 billion parameters allocated to the MLP layers. Since the original training data is not publicly available, we use a proprietary dataset consisting of high-quality pretraining data. With 4 choices of $b_j$, we sample different tokens for different budget goals (see details in Equ. 8), and use 60.4 billion training tokens in total.

**Downstream Tasks** We evaluate LLAMAFLEX on several downstream tasks including ARC-easy (Clark et al., 2018), LAMBADA (Paperno et al., 2016), PIQA (Bisk et al., 2020), WinoGrande (Sakaguchi et al., 2021), MMLU (Hendrycks et al., 2020), and HellaSwag (Zellers et al., 2019). Following the approach of (Xia et al., 2023), we report 5-shot performance for MMLU and 10-shot performance for HellaSwag, while presenting zero-shot results for the other tasks.

**Elastic Configurations and Training Details** As described in Section 2.2, for every budget target $b_j \in \mathcal{B}$, architectural choices, $D^j$, $N_A^j$, and $H^j$ are sampled from the predefined sets $\mathcal{D}$, $\mathcal{N}_A$, and $\mathcal{H}$, respectively. We present these sets of choices in Table 1. We omit the

Table 1: Details of elastic configurations.

| | | |
|---|---|---|
| Budget | $\mathcal{B}$ | $\{25\%, 50\%, 75\%, 100\%\}$ |
| MHA # Heads | $\mathcal{N}_A$ | $\{25\%, 50\%, 75\%, 100\%\}$ |
| MLP inter. dim | $\mathcal{D}$ | $\{25\%, 37.5\%, 50\%, 62.5\%, 75\%, 87.5\%, 100\%\}$ |
| Hid. dim | $\mathcal{H}$ | $\{50\%, 62.5\%, 75\%, 87.5\%, 100\%\}$ |

details of layer skipping, as its indicator variable $\lambda_i^j$ is a binary scalar that controls whether layer $i$ in model $j$ is skipped. Unless otherwise specified, we set the sequence length to 4096 and the batch size to 128, and

fine-tune the model for 28800 iterations. We set the initial learning rate to $4e - 5$, and use cosine learning rate decay for LLM parameters.

Table 2: Downstream task evaluation of LLAMAFLEX framework. Here, #Params refers to the number of *non-embedding* parameters. The prefix Exp. indicates that the sub-models are explicitly trained, i.e., $b \in \mathcal{B}$, while prefix Inter. refers to models generated through router interpolation.

| Type | Model | #Params | ARC-E | LAMB. | PIQA | Wino. | Hell. (10) | MMLU (5) | Avg. |
|------|-------|---------|-------|-------|------|-------|-----------|----------|------|
| LLAMAFLEX | Exp.-25% | 1.8 B | 66.7% | 61.6% | 74.3% | 61.9% | 67.6% | 42.2% | 62.4% |
| | Inter.-38.8% | 2.7 B | 71.3% | 64.6% | 76.0% | 62.5% | 71.1% | 53.8% | 66.6% |
| | Exp.-50% | 3.5 B | 76.1% | 66.3% | 77.0% | 67.9% | 76.0% | 60.3% | 70.6% |
| | Inter.-62.5% | 4.4 B | 77.4% | 68.1% | 77.4% | 69.4% | 76.4% | 61.2% | 71.7% |
| | Exp.-75% | 5.2 B | 78.9% | 71.7% | 78.1% | 72.9% | 77.2% | 62.7% | 73.6% |
| | Inter.-86.1% | 6.0 B | 79.2% | 72.1% | 78.8% | 74.1% | 77.9% | 63.1% | 74.2% |
| | Exp.-100% | 7.0 B | 81.9% | 72.8% | 79.7% | 75.4% | 78.0% | 64.7% | 75.4% |
| Flextron | 0.5× | 3.5 B | 65.9% | 61.7% | 74.8% | 61.9% | 67.6% | 35.9% | 61.3% |
| | 0.6× | 4.0 B | 66.1% | 63.8% | 75.0% | 62.1% | 68.0% | 37.7% | 62.1% |
| | 0.7× | 4.2 B | 65.8% | 64.2% | 75.6% | 62.3% | 67.1% | 41.9% | 62.8% |
| | Full | 6.5 B | 75.1% | 71.5% | 77.5% | 69.1% | 78.1% | 45.1% | 69.4% |
| Compression | Minitron-Depth | 3.7 B | 74.7% | 61.5% | 74.9% | 72.5% | 73.2% | 58.7% | 69.3% |
| | Minitron-Width | 3.7 B | 75.1% | 63.0% | 75.0% | 73.1% | 76.1% | 60.5% | 70.5% |
| | Sheared.-1.3B | 1.2 B | 61.5% | 61.0% | 73.4% | 57.9% | 60.7% | 25.7% | 56.7% |
| | Sheared.-2.7B | 2.5 B | 67.0% | 68.4% | 75.8% | 64.2% | 70.8% | 26.4% | 62.1% |
| | NutePrune | 3.2 B | 51.7% | - | 71.0% | 57.5% | 55.9% | - | - |
| | LLM-Pruner | 4.5 B | 59.2% | - | 73.4% | 64.2% | 56.5% | 23.9% | - |
| | Compresso | 4.5 B | 66.0% | - | 72.9% | 63.4% | - | 25.9% | - |
| | SliceGPT | 4.8 B | - | - | 66.2% | - | 50.3% | 28.9% | - |
| | LaCo | 4.7 B | - | - | 69.8% | - | 55.7% | 26.5% | - |
| Open-Source | Llama-3.1-8B | 7.0 B | 81.8% | 72.9% | 81.0% | 75.7% | 81.8% | 65.3% | 76.4% |
| | Llama2-7B | 6.5 B | 75.2% | 68.2% | 78.8% | 69.2% | 78.6% | 45.3% | 69.2% |
| | OpenLLaMA-3Bv2 | 3.2 B | 63.7% | 59.1% | 78.1% | 63.3% | 71.6% | 25.7% | 60.3% |
| | OpenLLaMA-7Bv2 | 6.5 B | 69.5% | 63.8% | 79.9% | 66.0% | 76.6% | 40.4% | 66.0% |
| | GPT3-8B | 6.4 B | 70.1% | 70.5% | 79.7% | 69.8% | 77.7% | 40.2% | 68.0% |
| | Pythia-1.4B | 1.2 B | 53.9% | 46.8% | 70.6% | 57.1% | 52.2% | 25.6% | 51.0% |
| | Pythia-2.8B | 2.5 B | 57.9% | 50.1% | 73.8% | 58.6% | 60.0% | 26.8% | 54.5% |
| | Pythia-6.9B | 6.4 B | 60.2% | 47.1% | 75.2% | 59.9% | 64.4% | 25.5% | 55.4% |

**Router Architecture and Modulation Details** For each architectural variable, we use a two-layer MLP followed by a shared embedding layer. The embedding layer converts the scalar $b_j$ into an embedding vector of dimension 128, and is shared across all architectural choices. Each MLP processes an intermediate dimension of 128 and outputs a logits vector, with a dimension corresponding to the number of elastic choices. The routers contains 0.51 million parameters in total. We use Gumbel-Softmax to optimize the routers, following the practice outlined in MaskLLM Fang et al. (2024). Specifically, we exponentially decay the temperature with rate 0.9999, and linearly scale the scaling factor $\kappa$ from 1 to 10. We set the initial learning rate of $4e - 2$ for router tuning. During the tuning process, we employ a combination of both soft and hard Gumbel-Softmax techniques (Jang et al., 2016). For each elastic choice, we modulate the elastic output by a sinusoidal embedding, of size 16, followed by a learnable lightweight MLP with an intermediate dimension of 128. Every modulation network only contains 0.02 million parameters. We also set the initial learning rate of $4e - 2$ for modulation networks.

## 3.2 DOWNSTREAM TASK EVALUATION

We present our downstream task evaluation results in Table 2. Here, we compare LLAMAFLEX against existing flexible inference frameworks such as Flextron (Cai et al., 2024), as well as open-source models, including Llama 3.1 8B (Dubey et al., 2024), Llama2 7B (Touvron et al., 2023), Pythia (Biderman et al.,

2023), OpenLLaMA (Geng & Liu, 2023), and GPT3 8B (Shoeybi et al., 2019). Additionally, we compare with models generated by post-hoc compression methods, including Minitron (Muralidharan et al., 2024), Sheared-LLaMA (Xia et al., 2023), Compresso (Guo et al., 2023), LLM-Pruner (Ma et al., 2023), SliceGPT (Ashkboos et al., 2024), and LaCo (Yang et al., 2024).

Since MMLU (Hendrycks et al., 2020) is a comprehensive benchmark for evaluating general knowledge across a wide range of domains, we provide a detailed visualization of the model's performance on benchmark in Figure 1. From Table 2 and Figure 1, we notice that LLAMAFLEX achieves the best performance-efficiency trade-off compared to baseline methods. In addition, the interpolated LLAMAFLEX models (denoted by the prefix `Inter.`) demonstrate a smooth transition in the performance-efficiency trade-off compared to the models explicitly receiving gradients (denoted by the prefix `Exp.`), highlighting the generalizability of our routers.

### 3.3 ROUTER VISUALIZATION

In Table 3, we visualize the 4 sub-networks learned by the LLAMAFLEX router; here, each sub-network satisfies a unique budget requirement $b_j$. As shown in the table, for higher budgets (e.g., 50% or 75%), the sub-networks generally avoid skipping layers and utilize relatively smaller MHA and MLP blocks. However, for lower budgets (e.g., 25%), the router tends to skip nearly half of the layers. Additionally, we observe that the router finds it hard to reduce the number of attention heads for Llama 3.1, and it remains the same for all different sub-

Table 3: Architecture details of sub-networks selected by the learnable router. Each sub-network corresponds to a different budget $b_j$; the architecture of the pre-trained Llama 3.1 8B model is shown in the $b_j = 100\%$ row. We present the MLP intermediate dimension $D^j$, the number of attention heads (the number of queries) $N_A^j$, the hidden dimension $H^j$, the number of layers $N^j$, and the resulting number of non-embedding parameters. Note that since the pretrained model uses grouped-query attention and we keep the number of attention groups fixed, we modify the number of queries per group.

| $b_j$ | #Param | MLP Dim. | MHA #Heads | Hid. Dim | #Remain. layers |
|---|---|---|---|---|---|
| 25% | 1.84$B$ | 10752 (75.0%) | 24 (75%) | 2560 (62.5%) | 18 (56.25%) |
| 50% | 3.48$B$ | 12544 (87.5%) | 24 (75%) | 2560 (62.5%) | 30 (93.75%) |
| 75% | 5.20$B$ | 12544 (87.5%) | 24 (75%) | 3584 (87.5%) | 32 (100%) |
| 100% | 6.98$B$ | 14336 | 32 | 4096 | 32 |

networks. For the 25% variant, layers $[1-7, 17-24, 29-30, 32]$ are retained while the others are skipped. In the 50% variant, the 13-th and 31-th layers are removed.

## 4 ABLATION STUDY

**Effect of Policy-Aware Modulation** We validate the effectiveness of our modulation technique by comparing performance with and without modulation. We set $\mathcal{B} = \{25\%, 50\%, 75\%, 100\%\}$, and perform LLAMAFLEX training with and without modulation. We train Llama

Table 4: Results of our ablation study on policy-aware modulation. We run LLAMAFLEX for 800 iterations and report the validation loss. All sub-networks show improved performance when modulation is enabled, reducing validation loss by 0.08 on average.

| $b_j$ | 25% | 50% | 75% | 100% | avg. |
|---|---|---|---|---|---|
| w/ Modulation | 2.53 ($\downarrow$ 0.08) | 2.41 ($\downarrow$ 0.13) | 2.08 ($\downarrow$ 0.09) | 1.88 | 2.22 ($\downarrow$ 0.08) |
| wo Modulation | 2.61 | 2.54 | 2.17 | 1.88 | 2.30 |

3.1 8B (Dubey et al., 2024) for 800 iterations, and show the obtained performance in Table 4. We notice that including modulation consistently improves the performance of all sub-networks, reducing validation loss by 0.08 on average. We provide more experimental results in Appendix A.

## 5 RELATED WORK

**Structured Pruning** A number of recent structured pruning papers specifically target LLMs; we can broadly classify these works into two main categories: (1) ones that prune only depth (layers), and (2)

ones that prune width (attention heads, MLP intermediate dimension, etc.) and/or depth. Recent work in the first category (depth pruning) includes ShortGPT Men et al. (2024), LaCo Yang et al. (2024), and Shortened LLaMa Kim et al. (2024), while those in the second category includes Minitron Muralidharan et al. (2024), ShearedLlama Xia et al. (2023), Dery et al. (2024), SliceGPT Ashkboos et al. (2024), and LLM-Pruner Ma et al. (2023). To the best of our knowledge, all the work in this area requires fine-tuning or distillation on a per-compressed-model basis (not zero-shot). We compare the accuracy of LLAMAFLEX models against a large subset of these frameworks in this paper.

**Flexible Architectures**    Flexible inference has been extensively studied, particularly in the context of convolutional neural networks (CNNs). Yu et al. (2018); Yu & Huang (2019) introduced slimmable neural networks, which enabled the deployment of the same model with varying numbers of convolutional kernels. Building on this, OFA (Cai et al., 2019) generalized pruning techniques to create a single model that can adapt to multiple configurations. Recent research has extended slimmable models to Transformer architectures. Kusupati et al. (2022) introduced a nested weight structure for Transformer networks, focusing primarily on hidden features. Matformer (Kudugunta et al., 2023) further developed this architecture by applying it to the intermediate dimensions of MLPs. Additionally, SortedNet (Valipour et al., 2023) employed a sampling-based training strategy to train multiple models via gradient accumulation, while SortedLlama (Kavehzadeh et al., 2023) explored depth-wise sampling in large language models (LLMs). Recently, Flextron (Cai et al., 2024) introduced elastic multi-head attention (MHA) and search-based routing to enhance flexibility. Similar to Flextron, LLAMAFLEX leverages a router mechanism, but supports end-to-end training, interpolation between trained targets, and produces more easily deployable uniform architectures. Additionally, LLAMAFLEX introduces a novel policy-aware modulation technique, providing additional model expressivity and enhancing adaptability.

**SuperNet-based Neural Architecture Search (NAS)**    HAT (Wang et al., 2020) and HELP (Lee et al., 2021) have explored supernet-based approaches for generating sub-networks, primarily targeting relatively small-sized models, while Meta-NAS (Elsken et al., 2020) and DARTS-EGS (Chang et al., 2019) have utilized Gumbel-Softmax-based supernet techniques. Our work focuses on addressing the unique challenges of LLMs, which differ significantly from prior studies targeting smaller models. To avoid costly training processes and huge GPU memory usage, we transform a pre-trained model into an elastic framework, and employ weight-sharing, unlike previous attempts. To avoid repeated computation, we enable zero-shot subnetwork generation for any parameter budget through router interpolation - capabilities not achieved by supernet-based methods.

## 6    CONCLUSIONS

In this paper, we have presented LLAMAFLEX, a novel elastic LLM architecture that supports zero-shot resizing along both width and depth dimensions, yielding a large space of uniform (for ease of deployment) compressed models without any fine-tuning. It utilizes a Gumbel Softmax-based end-to-end learnable router that requires a one-time training phase, and is then able to smoothly interpolate across model sizes, ranging from 0 to 100% of the original model size. We have also introduced a novel policy-aware modulation technique that improves the generality of nested architectures. LLAMAFLEX on Llama 3.1 8B produces a frontier of highly accurate compressed models that outperform similarly-sized compressed, elastic/flexible, and trained-from-scratch models.

**Limitations.**    Our method is not training-free, as it requires router tuning and necessitate modification on backbone weights to convert the pre-trained model into an elastic one. There is a potential application of training LLAMAFLEX from scratch, we leave this to future work.

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

## A MORE EXPERIMENTAL RESULTS

**Performance of Learnable Layer Skipping** All existing techniques for layer skipping that we are aware of use hand-crafted heuristics to measure the importance of layers and subsequently prune the layers with the lowest importance scores (Men et al., 2024; Gromov et al., 2024; Siddiqui et al., 2024; Muralidharan et al., 2024). In contrast, our proposed method uses a Gumbel Softmax-based router to optimize layer skipping. To evaluate how well our approach works, we use the full transformer block and only enable layer skipping; thus, to produce a sub-network with 50% remaining parameters we skip half the transformer layers. We compare against two baselines: (1) random layer skipping, and (2) skipping the last half of the layers, except for the final layer (i.e., layers 15 to 31 in 32-layers Llama 3.1 8B model), which follows the depth-pruning approach used by Muralidharan et al. (2024). We train the models for 1600 iterations and present the evaluation loss in Figure 5. We observe that while heuristic skipping converges more quickly than other methods, our approach demonstrates superior performance once training has converged.

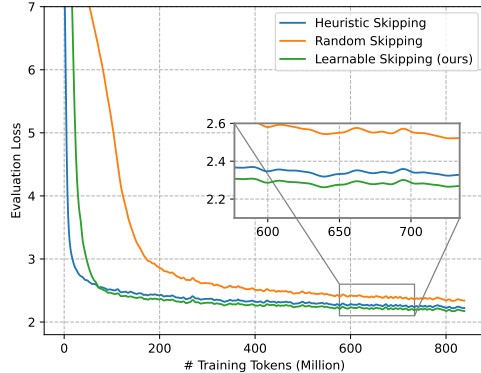

Figure 5: Evaluation of the proposed learnable layer skipping method. Here, we use the full Transformer block and only enable layer skipping in the router, setting it to use 50% of layers. Our method (green curve) is compared against random skipping (orange curve) and the approach used by Minitron (Muralidharan et al., 2024) (blue curve).

