# OpenReview forum: "LLaMaFlex: Many-in-one LLMs via Generalized Pruning and Weight Sharing"
_ICLR.cc/2025/Conference — ICLR 2025 Poster_

### Official Review · Reviewer_gENy · 2024-11-03

**Soundness:** 3
**Presentation:** 2
**Contribution:** 3
**Rating:** 6
**Confidence:** 4

**Summary:**

The paper presents LLaMaFlex, an elastic architecture for large language models that enables zero-shot resizing across both width and depth dimensions, allowing for the instant generation of multiple compressed models without additional fine-tuning. LLaMaFlex utilizes a nested weight-shared network architecture and a Gumbel Softmax-based router for smooth interpolation between model sizes, achieving a "train once, deploy many" paradigm. It introduces policy-aware modulation to enhance the expressivity of nested architectures and produces a family of compressed models from Llama 3.1 8B that outperform state-of-the-art compressed, elastic/flexible, and trained-from-scratch models in accuracy. The uniform architectures generated by LLaMaFlex are also easier to deploy using existing LLM frameworks, offering a significant advancement in training efficiency and deployment flexibility for large language models.

**Strengths:**

1. The authors introduce a novel elastic architecture for large language models that enables zero-shot resizing across both width and depth, addressing a key challenge for efficient model deployment.

2. The end-to-end trained router could dynamically determine the optimal network architecture under a given parameter budget.

3. It achieves a promising performance of the compressed model in various of size, which outperform the same size model trained from scratch.

**Weaknesses:**

1. The authors validate their proposed method in only one model, Llama 3.1 8B. The serviceability to other models and architecture should be considered.

2. I understand that the router is used to determine the optimal architecture, that is the hyperparams of the network architecture, under a given parameter budget. Once the hyperparams of the network architecture are determined, the algorithm will select the top-k importance submodule, which is ranked by a pre-defined metric, e.g. the accumulated magnitude of activations as stated in Section 2.3. There is no ablation on the pre-defined metric, which might limit the rigor and integrity of this work.

3. How to guarantee that the sampled submodel (i.e., the combination of N, H, D, N_A, lambda) satisfies the parameter budget (i.e., the second equation in Eq. 4), given that N, H, D, N_A, and lambda are modeled independently in the routers (i.e., Eq. 5)? What is the explicit joint distribution of Q in Eq. 6?

4. The authors did not explicitly discuss the limitations of the proposed method.

**Questions:**

1. Instead of interpolating the router input h_k in Eq. 7, is it possible to interpolate the router output in a similar way given h_{n+1} and h_{n}?

2. In Line 196, the author indicates that \mathcal{D}, \mathcal{N_A}, \mathcal{N}, \mathcal{H} are predefined, how to define them?

---

> ### Author Response · Authors · 2024-11-22
>
> **[W1: Extend to other Pre-trained Model Type]**
> Thank you for the suggestion. We validated our method on Minitron-4B [1], a representative efficient large language model of relatively small scale. Using the same hyperparameter settings as in our original experiments on Llama-3.1-8B, we trained the model for 10k iterations and evaluated the perplexity (PPL) on four resultant sub-networks.
>
> Table D1: Evaluation of our method using Minitron-4B as the pre-training model. We report the anchor model (the sub-models directly received gradient during training), as well as the interpolated model (the sub-models generated by router interpolation).
> | Model size | 25% (anchor) | 37.5% |  50% (anchor)  | 62.5% | 75% (anchor)  | 87.5% | 100% (anchor)  |
> |:-:|:-:|:-:|:-:|:-:|:-:|:-:|:-:|
> | PPL | 25.25 | 22.89 | 21.22 | 20.03 | 17.42 | 14.89 | 11.78 |
>
> **[W2: Clarify our approach, and Metric Ablation]**
> Thank you for the suggestion. Using the proposed importance metric (cumulative activated values), we rank the channels and permute the weight matrix to reorder the channels/heads in decreasing order of importance. To ablate the metric, we compare this ranking with permutation to a setup without permutation. After training the model for 2k iterations, we observe a significant improvement in validation loss when incorporating the importance-based metric.
>
> Table D2: Ablation experiments on our proposed importance metric.
> | Model size | 25% |  50%  | 75% | 100% |
> |:-:|:-:|:-:|:-:|:-:|
> | wo Ranking & Permutation | 2.89 | 2.53 | 2.25 | 2.01 |
> | w/ Ranking & Permutation | 2.58 | 2.37 | 2.18 | 2.01 |
>
> It is worth noting that our importance metric builds upon previous work. Specifically, both the Minitron[1] and Flextron[2] papers include detailed ablations comparing various metrics for importance estimation (see Sections 4.2 and 5.5 in the Minitron and Flextron papers, respectively). We use the best metrics proposed in these works.
>
> **[W3: sampled network]**
> Thank you for the question. As detailed in Equation 4, our overall training objective is to minimize the loss of the sampled sub-networks while ensuring they satisfy the parameter budget constraints. This is achieved by incorporating an additional loss term that is activated only when the actual number of activated parameters exceeds the budget. This approach is standard and has been adopted in existing works [2]. Further details are provided in Section 2.3. In terms of joint distribution of Q, we independently model the categorical distribution of each architectural choice by a MLP and sample one category with gumbel softmax. Specifically, $Q(D^j, N_A^j, H^j, N^j, \boldsymbol{\lambda}^j) = P(D^j) P(N_A^j) P(H^j) P(N^j) \prod_i P(\lambda^j_i)$.
>
> **[W4: Limitation discussion]**
> Thanks for your suggestion. We have included a limitations section in the revised version (marked in blue) as below:
>
> Our method is not training free, as it requires router tuning and necessitates modification to backbone weights to convert the pre-trained model into an elastic one. There is a potential application of training LlamaFlex from scratch; we leave this to future work.
>
> **[Q1: Router interpolation]** Thank you for the interesting question. The router outputs cannot be interpolated because they are designed to model the categorical distribution of each architectural choice. The model samples one architecture choice from this distribution, and as training progresses, the sampling process becomes increasingly deterministic, consistently selecting the optimal choice. This results in the router outputs converging to a nearly one-hot distribution. Interpolating the router outputs would disrupt this one-hot distribution, undermining the intended selection mechanism.
>
> **[Q2: Configuration details]**
> Thanks for the question. We provide the choices of \mathcal{D}, \mathcal{N_A}, \mathcal{N}, \mathcal{H} in Table 1 in Section 3.1. Specifically:
> - \mathcal{N}_A = \{25\%, 50\%, 75\%, 100\%\}
> - \mathcal{D} = \{25\%, 37.5\%, 50\%, 62.5\%, 75\%, 87.5\% , 100\%\}
> - \mathcal{H} = \{50\%, 62.5\%, 75\%, 87.5\% , 100\%\}
>
> We observe that the final results are largely independent of specific choices within these sets; in other words, the selections can be made flexibly, as long as there exist combinations of these sets that satisfy the parameter budget constraints and the elements within each set are reasonably spaced.
>
> [1] Compact language models via pruning and knowledge distillation
>
> [2] Flextron: Many-in-One Flexible Large Language Model

---

> > ### Author Response · Authors · 2024-11-29
> >
> > Thank you very much for your thorough review and for the thoughtful comments you provided on our work.
> >
> > We have made the necessary revisions in response to your comments, and we hope that our revisions meet your expectations. If there are any remaining concerns or if you have any further suggestions that could help us improve the quality of our work, please do not hesitate to let us know. We would greatly appreciate any further guidance you can provide.
> >
> > Once again, we are truly grateful for your support.
> >
> > Authors

---

> ### Comment · Area_Chair_gXjG · 2024-11-30
>
> Dear Reviewer,
>
> Could you kindly respond and indicate whether authors have addressed your concerns?
>
> Thanks, AC

---

> ### Comment · Reviewer_gENy · 2024-12-02
>
> Dear authors, thank you for your thorough responses, which addressed my concerns, therefore I raise the score to 6.

---

> > ### Author Response · Authors · 2024-12-02
> >
> > Many thanks for your thoughtful comments and suggestions. We value your support and will be incorporating all your feedback in our revised version.

---

### Official Review · Reviewer_XXXs · 2024-11-04

**Soundness:** 3
**Presentation:** 3
**Contribution:** 3
**Rating:** 6
**Confidence:** 3

**Summary:**

This paper presents LLaMaFlex, a novel approach to create elastic Large Language Models (LLMs) that can be dynamically resized without additional fine-tuning. The key innovation is a nested weight-shared architecture combined with a Gumbel Softmax-based router that allows zero-shot pruning across both width (attention heads, hidden dimensions, MLP intermediate size) and depth (layers) dimensions. Unlike previous approaches that require separate training or distillation for each model size, LLaMaFlex needs only a single continued training phase of ~60B tokens to enable "train once, deploy many" capabilities. The authors also introduce a policy-aware modulation technique inspired by diffusion models to enhance the expressiveness of nested architectures. When applied to Llama 3.1 8B, LLaMaFlex produces compressed models that outperform state-of-the-art pruned models, elastic frameworks, and models trained from scratch. The router can interpolate smoothly between different model sizes, allowing deployment flexibility without accuracy compromises.

**Strengths:**

- The paper introduces several innovative technical components, including an end-to-end learnable router with Gumbel Softmax for architecture selection, policy-aware modulation for enhanced expressiveness, and support for both width and depth pruning dimensions. This comprehensive approach advances the state-of-the-art in elastic LLM architectures.
- The method demonstrates impressive performance across multiple downstream tasks (ARC-E, LAMBADA, PIQA, WinoGrande, HellaSwag, MMLU), consistently outperforming existing approaches including Flextron, Minitron, and models trained from scratch. The results are particularly notable given that LLaMaFlex requires only 60B tokens for training compared to hundreds of billions for traditional approaches.
- LLaMaFlex produces uniform architectures that are compatible with common deployment frameworks like TensorRT-LLM and llama.cpp, addressing a significant limitation of previous elastic approaches that generated heterogeneous architectures. The ability to interpolate between model sizes without additional training also offers practical deployment flexibility.

**Weaknesses:**

- While the paper presents strong technical contributions, there are opportunities for additional context and analysis that could further strengthen the work. The authors could enrich their discussion by connecting their approach to the broader context of supernet-based neural architecture search. For instance, works like HAT and HELP have previously explored supernet-based approaches for generating target-budget subnets in the field of small language models and Meta-NAS or DARTS-EGS have handled Gumbel-Softmax-based supernet. Additionally, a more thorough discussion and comparison of the paper's relationship to MatFormer's subnet extraction approach for Transformer architectures (applied to Llama or ViT) would be valuable.
- The paper would benefit from additional analysis of training dynamics typically associated with supernet approaches. In supernet-based methods, there are well-known challenges to address. First, training imbalances between larger and smaller subnets often occur, where larger architectures receive less training attention and may not fully converge. Second, weight sharing interference can arise when different architectural configurations compete for optimal parameter values. While the authors present strong results, providing insights into how LLaMaFlex addresses these potential training imbalances would be informative. Similarly, a discussion of how weight sharing affects different architectural configurations could offer valuable implementation insights for practitioners.
- The comparative analysis could be more comprehensive, particularly regarding scenarios with fixed target sizes. A direct comparison with knowledge distillation approaches would be valuable, considering factors such as training computational costs, final model performance, and the associated trade-offs. Such analysis would help practitioners better understand when to choose LLaMaFlex over traditional approaches.
- The process for selecting optimal architectures when multiple configurations achieve similar parameter counts needs more detailed explanation.

[HAT] HAT: Hardware-Aware Transformers for Efficient Natural Language Processing
[HELP] HELP: Hardware-Adaptive Efficient Latency Predictor for NAS via Meta-Learning
[MatFormer] MatFormer: Nested Transformer for Elastic Inference
[MetaNAS] Meta-Learning of Neural Architectures for Few-Shot Learning
[DARTS-EGS] Differentiable Architecture Search with Ensemble Gumbel-Softmax

**Questions:**

Please address concerns in Weaknesses.

---

> ### Author Response · Authors · 2024-11-22
>
> **[W1: Comparison with Supernet-based NAS]**
> Thanks for the insightful suggestion. Our work primarily focuses on LLMs, which sets it apart from previous studies that primarily target relatively small models. Consequently, our work offers unique contributions specifically designed to address the challenges associated with LLMs:
> - LLMs demand substantial computational resources and extensive datasets for training. To mitigate these high costs, our framework focuses on **transforming a pre-trained model** into an elastic framework, enabling efficient training and adaptation. In contrast, supernet-based methods such as HAT [1], HELP [2], DARTS-EGS [3] and MetaNAS [4], as well as recent work like Matformer [5], primarily rely on training models from scratch. Additionally, our method enables zero-shot generation of sub-networks for any parameter budget through router interpolation, a capability not achieved by previous supernet-based methods.
> - LLMs typically consist of billions of parameters, making GPU memory usage a critical bottleneck during training. To address this, we adopt weight sharing, ensuring the number of training parameters remains comparable to that of a dense pre-trained model. In contrast, methods like HELP [2], DARTS-EGS [3], and MetaNAS [4] train a supernet with multiple branches. While HAT [1] also employs weight sharing, its use of heterogeneous layers complicates system deployment.
> - Matformer [5] relies on manual mix-and-match to extract sub-models, requiring manual selection of optimal configurations. In contrast, we present a fully learnable router that can do this sub-model extraction automatically.
>
> We have added the following paragraph to the related work section in the revised PDF:
>
> HAT[1] and HELP[2] have explored supernet-based approaches for generating sub-networks, primarily targeting relatively small-sized models, while Meta-NAS[4] and DARTS-EGS[3] have utilized Gumbel-Softmax-based supernet techniques. Our work focuses on addressing the unique challenges of LLMs, which differ significantly from prior studies targeting smaller models. To avoid costly training processes and huge GPU memory usage, we transform a pre-trained model into an elastic framework, and employ weight-sharing, unlike previous attempts. To avoid repeated computation, we enable zero-shot sub-network generation for any parameter budget through router interpolation - capabilities not achieved by supernet-based methods.
>
>
> **[W2: Training dynamics]**
> Our framework leverages **pre-trained LLMs** and applies weight matrix permutation based on an importance metric, ensuring that the largest model variant (i.e., the full pre-trained model) is already converged while other sub-models are reasonably initialized. Furthermore, the algorithm inherently avoids selecting the converged large model variant, as it is penalized by the parameter loss for exceeding the budget requirement.
>
> To mitigate the interference caused by weight sharing, we propose the **modulation** technique (detailed in Sec 2.4). Specifically, the nesting constraint can potentially limit the representational capacity of elastic networks, as the same set of weights must accommodate inputs across all possible sub-networks. To address this limitation, we propose the use of policy-aware modulation, a technique inspired by methods in the literature on diffusion models. We introduce lightweight non-linear modulation heads, which are applied after the elastic components (i.e., elastic MLP/elastic MHA). These heads modulate the outputs of elastic operations based on the elastic choice. We ablate the modulation technique in Section 4 and provide a detailed comparison in Table C1 below.
>
> Table C1: Results of our ablation study on policy-aware modulation. We run LLAMAFLEX for 800 iterations and report the validation loss. All sub-networks show improved performance when modulation is enabled, reducing validation loss by 0.08 on average.
> |$b_j$|25%|50%|75%|100%|avg.|
> |:-:|:-:|:-:|:-:|:-:|:-:|
> |w/ Modulation|2.53 ($\downarrow$ 0.08)|2.41 ($\downarrow$ 0.13)|2.08 ($\downarrow$ 0.09)|1.88|2.22 ($\downarrow$ 0.08)|
> |wo Modulation|2.61|2.54|2.17|1.88|2.30|
>
> [1] HAT: Hardware-Aware Transformers for Efficient Natural Language Processing
>
> [2] HELP: Hardware-Adaptive Efficient Latency Predictor for NAS via Meta-Learning
>
> [3] Differentiable Architecture Search with Ensemble Gumbel-Softmax
>
> [4] Meta-Learning of Neural Architectures for Few-Shot Learning
>
> [5] MatFormer: Nested Transformer for Elastic Inference

---

> ### Author Response · Authors · 2024-11-22
>
> **[W3: Comparison with Distillation]**
> Thank you for your insightful question. First, we would like to emphasize that elastic frameworks, including LlamaFlex, are orthogonal to traditional fixed-size compression methods. While our primary focus is on the elastic component, our framework is fully compatible with various fixed-size compression approaches and can potentially benefit from advanced methods in this domain. To validate this compatibility, we incorporated knowledge distillation into our framework by adding an additional knowledge distillation loss. We trained the model for 2k iterations and reported the validation loss in Table C2 below. This experiment demonstrates the flexibility of our approach in integrating with other compression techniques.
>
> Table C2: Compatibility of the proposed framework with existing fix-sized compression methods.
> | Model size| 25% | 50% | 75% | 100%|
> |:-:|:-:|:-:|:-:|:-:|
> | wo distillation | 2.58 | 2.35 | 2.16 | 1.97 |
> | w/ distillation | 2.49 | 2.26 | 2.15 | 1.99 |
>
> Second, we compared our method with representative fixed size compression methods - Minitron [1]. To ensure fair comparison, we use the same pre-trained model and similar data sources as Minitron, and compare the downstream performance in Table C3.
>
> Table C3: Downstream comparison with Minitron method (summarized from Table 2).
> | Model | #Params | ARC-E | LAMB. | PIQA | Wino. | Hell. | MMLU | Avg.|
> |:-:|:-:|:-:|:-:|:-:|:-:|:-:|:-:|:-:|
> | LlamaFlex-50% | 3.5B | 76.1% | 66.3% | 77.0% | 67.9% | 76.0% | 60.3% | 70.6% |
> | Minitron-Depth | 3.7 B | 74.7% | 61.5% | 74.9% | 72.5% | 73.2% | 58.7% | 69.3% |
> | Minitron-Width |  3.7 B| 75.1%| 63.0% | 75.0%| 73.1%| 76.1%| 60.5%| 70.5%|
>
> **[W4: Discussion on optimal architectures]**
> Thank you for the suggestion! In our framework, a lightweight sub-network can be derived by reducing the hidden dimension ($H$), MLP intermediate dimension ($D$), number of attention heads ($N_A$), and the number of remaining layers ($N$). This framework offers a wide range of possible combinations to achieve approximately the same parameter budget (e.g. 50% remaining parameters). However, different configurations would yield different performance. Thus we apply the learnable framework that aims to search for the sub-network with optimal configurations.
>
> Table C4: The learnable framework yields optimal sub-network configurations. With a 50% remaining parameter budget, the sub-network can adopt diverse configurations. For example, we can prune the MLP by retaining only half of its intermediate dimension, prune the hidden dimension by keeping only half of the hidden feature size, or skip half of the layers to meet the 50% target. However, our learnable framework identifies the optimal configurations that combine multiple strategies. All models are fine-tuned for 2k iterations in these experiments.
>
> |Method | $H$ | $D$ | $N_A$ | $N$ | Budget (#param) | Valid Loss |
> |:-:|:-:|:-:|:-:|:-:|:-:|:-:|
> | MLP pruning-only | 4096 | 7168 (50%) | 32 | 32 | 4.16 B (59%) | 2.16 |
> | Hidden-dim pruning-only | 2048 (50%) | 14336 | 32 | 32 | 3.49 B (50%) | 2.22 |
> | Layer pruning-only | 4096 | 14336 | 32 | 16 (50%) | 3.49 B (50%) | 2.31 |
> | Learnable (ours) | 2560 (62.5%) | 12544 (87.5%) | 24 (75%) | 30 (93.75%) | 3.48 B (50%) | 2.12 |
>
> [1] Compact language models via pruning and knowledge distillation

---

> > ### Author Response · Authors · 2024-11-29
> >
> > Thank you very much for your thorough review and for the thoughtful comments you provided on our work.
> >
> > We have made the necessary revisions in response to your comments, and we hope that our revisions meet your expectations. If there are any remaining concerns or if you have any further suggestions that could help us improve the quality of our work, please do not hesitate to let us know. We would greatly appreciate any further guidance you can provide.
> >
> > Once again, we are truly grateful for your support.
> >
> > Authors

---

> ### Comment · Area_Chair_gXjG · 2024-11-30
>
> Dear Reviewer,
>
> Could you kindly respond and indicate whether authors have addressed your concerns?
>
> Thanks, AC

---

### Official Review · Reviewer_uPJk · 2024-11-04

**Soundness:** 3
**Presentation:** 1
**Contribution:** 2
**Rating:** 6
**Confidence:** 5

**Summary:**

LLAMAFLEX introduces a new approach to efficiently and effectively generate high-accuracy SLMs. LLAMAFLEX employs a nested weight-shared architecture. Starting from a pre-trained model, it uses only 60 billion tokens for a single training phase, enabling zero-shot pruning across both width and depth and a “train once, deploy many” pruning paradigm.

**Strengths:**

+ Introduces a gumbel softmax-based end-to-end learnable router to train the candidate models.
+ Enhances generality by incorporating a policy-aware modulation technique.
+ Enables interpolation-based extrapolation of pruning rates for flexible model compression.
+ Achieves accuracy comparable to state-of-the-art SLMs like minitron.

**Weaknesses:**

- The training process and methodology flow are difficult to understand and a figure like Figure 3 in Flextron paper should be provided.
- The presentation in the main text is somewhat weaker than in the abstract, e.g. the explanation of Equation 3 (L170-L180), and the textual ordering needs to be organized. For another example, paragraphs L199-L210 are intended to be placed in the heading “Router Design” rather than under “Problem Formulation”. Apart from that, parts of 2.3 seem to fit into the experimental chapter. More, the description of the amount of data on line 325 should be placed on line 309 and the source of the data set should be described.
- In Eq. 2, if \lambda_i^j=0, it would be misleading to think that all previous layers have been pruned, and the formula should be reshaped to alleviate the ambiguity. The upper limit of j is not defined, and if j can be taken to infinity, this should be stated.
- Unfair comparison with Flextron, you should reproduce Flextron-Llama3.1-8B to provide a fair comparison. Unfair comparison with structured pruning methods: 1) All of these methods need to be reproduced with llama-3.1-8B as the base model. 2) Structured pruning methods almost always support fine-tuning after pruning, and you should implement training using the same amount of data as your work to achieve a fair comparison.
- While LLAMAFLEX can incubate SLMs similar to Minitron accuracy, the layer-pruning-only version of Minitron-LLaMA-3.1 will embody far more efficiency than an SLM of the same size, and you need to make a full comparison.

**Questions:**

- L094-L096: The value of a heterogeneous architecture should not be dismissed because of unfriendly backend support. I believe that loosening the homogeneous restrictions in LLAMAFLEX will boost the potential to produce stronger models at the same scale. Can you provide some relevant results (or insights if it is resource-intensive to build this experiment)?
- Are open-source SLMs in the Phi series and the LLaMA 3.2 series comparable in effectiveness and worth adding to the evaluation?
- How much computational and time resources were used for this work?

---

> ### Author Response · Authors · 2024-11-22
>
> **[W1: Pipeline Figure]**
> Thank you for the suggestion. We have updated Figure 3 to explicitly illustrate how the routers determine the sub-network configurations. Additionally, we have included a new Figure 4 to provide a clearer depiction of the modulation process.
>
> **[W2: Text and subtitles]**
> Thank you for your valuable suggestions. We have updated the revised PDF and marked the changes in **blue** based on your suggestions. Please let us know if there are any further questions or points of confusion. Regarding data resources, we utilize in-house data but are unfortunately unable to disclose specific details at this stage to avoid revealing author identities, in compliance with the double-blind review policy. More detailed information will be provided upon paper acceptance.
>
> **[W3: Confusion on Equation 2]**
> Thanks for pointing this out and our apologies for the confusion. We have updated Equations 1 and Equation 2 in the revised PDF, and have added the skip connection term in the second equation. In this case, the layer will be skipped and only the skip connection is enabled if $\lambda_i$ equals 0.
>
> **[W4: Unfair comparison, and the comparison with layer-pruning-only Minitron]**
> Thank you for pointing this out. Our method primarily compares with Minitron, a representative compression approach that integrates pruning and distillation. To ensure a fair comparison, we used Llama-3.1-8B as the pre-trained base model and the similar data distribution (confirmed with the Minitron authors), aligning with the setup of Minitron-4B to the best of our knowledge.
>
> As shown in Table B1 below, our method outperforms the Minitron models (both the layer-pruning and width-pruning versions). Notably, our approach requires only 60 billion tokens, compared to Minitron's 200 billion tokens (100 billion tokens per model version, with computations repeated). Despite using significantly fewer tokens, we notice that our approach achieves better performance, underscoring its effectiveness.
>
> Table B1: Downstream comparison with Minitron method (summarized from Table 2).
> | Model | #Params | ARC-E | LAMB. | PIQA | Wino. | Hell. | MMLU | Avg.|
> |:-:|:-:|:-:|:-:|:-:|:-:|:-:|:-:|:-:|
> | LlamaFlex-50% | 3.5B | 76.1% | 66.3% | 77.0% | 67.9% | 76.0% | 60.3% | 70.6% |
> | Minitron-Depth | 3.7 B | 74.7% | 61.5% | 74.9% | 72.5% | 73.2% | 58.7% | 69.3% |
> | Minitron-Width |  3.7 B| 75.1%| 63.0% | 75.0%| 73.1%| 76.1%| 60.5%| 70.5%|
>
> We acknowledge that the claim "LlamaFlex can incubate SLMs similar to Minitron accuracy" may not fully capture the nuances of the comparison. As shown in Table B1, LlamaFlex-50% achieves higher average accuracy (70.6%) compared to Minitron-Depth (69.3%) and is on par with Minitron-Width (70.5%). Moreover, LlamaFlex requires significantly fewer tokens for fine-tuning.
>
> **[Q1: Heterogeneous Layer]**
> Thank you for your insightful question. While our primary focus is on a system-friendly elastic framework, we observed the trade-off between performance and system efficiency. Loosening the homogeneous restrictions in LlamaFlex could enhance its potential to produce stronger models at the same scale. Specifically, we explored using layer-wise routers instead of a single router for all layers, and this approach demonstrated improved performance while maintaining the same model size.
>
> **[Q2: Comparison with Phi]**
> Thank you for the comments. Our model cannot be directly compared with the Phi series or Llama 3.2 due to differences in data resources. To ensure a fair comparison, we focus on benchmarking against the Minitron method, where our model demonstrates exceptional performance while utilizing fewer training resources.
>
> **[Q3: Training details]** Thank you for the question. We used a total of 60.4 billion training tokens and approximately 12,288 GPU hours. Specifically, we set the sequence length to 4096, the batch size to 128, and fine-tuned the model for 28,800 iterations.

---

> > ### Comment · Reviewer_uPJk · 2024-11-26
> > **Comment**
> >
> > Thank you for your feedback, and I recommend that you organize your newly provided experimental results in the main text or in an appendix. Methods such as Flextron and Sheared-llama that were used as comparisons should still be re-used with Llama-3.1 as the base model to ensure fair comparisons (you need not provide these comparisons at the rebuttal stage due to time limitations). Your reply solved my problem to a considerable extent and I would like to keep my score the same, thanks.

---

> > > ### Author Response · Authors · 2024-11-29
> > >
> > > Thank you for your thoughtful comments and suggestions. Due to the limited rebuttal period, we were unable to complete the full comparison. However, we greatly appreciate your support and will incorporate all your feedback into the revised version.

---

### Official Review · Reviewer_7Wyq · 2024-11-04

**Soundness:** 3
**Presentation:** 3
**Contribution:** 3
**Rating:** 8
**Confidence:** 4

**Summary:**

This paper introduces a novel LLM compression framework named LLaMaFlex. With around ~60B tokens for elastic training, the network can be resized both in the depth and the width dimension, producing better results than previous pruning or distillation-based methods. The method composes a Gumbel-softmax router to select the sub-network after the elastic training. For the training, the model is first arranged by the importance of sub-networks and a policy-aware modulation is proposed. Experiments show that it can achieve much better results than previous compression methods.

**Strengths:**

1. The idea of “train once, generate many” for LLM is interesting for compressing large language models. While this paradigm isn’t entirely new, this paper achieves impressive results to apply this on LLM.
2. The proposed method is both well-founded and novel. A particularly interesting discovery is that the learned router can interpolate, enhancing the method’s generalizability.
3. The experimental results demonstrate strong performance.

**Weaknesses:**

I did not find significant weaknesses for this paper

**Questions:**

1. How does the performance compare if elastic training is not applied to retrain the LLM? Specifically, if only the Gumbel router is learned on the pre-trained LLM, while retaining the pre-trained weights of LLaMA-3.1-8B.
2. Have you conducted experiments on other LLMs to demonstrate that this method can be extended across different types of language models?

---

> ### Author Response · Authors · 2024-11-22
>
> **[Q1: Learn routers while retaining pre-trained model weights]**
> Thank you for your insightful suggestion. Our method remains effective when retaining model weights, and the routers are able to search the best configurations of the sub-networks. We conduct an experiment where we fix the backbone weights and enable only the learnable routers to search for the optimal sub-network configurations under two parameter budgets: 25% and 50%. The results, compared against heuristic methods, are presented in Table A1 below.
>
> Table A1: Comparison of learnable searching framework (ours) with heuristic based methods, on elastic training while retaining the original weights.
> | Method | Valid Loss (50% #param) | Valid Loss (25% #param) |
> |:-:|:-:|:-:|
> | MLP pruning-only | 4.35| 7.14 |
> | Hidden Dim. pruning-only | 5.41 | 8.99 |
> | Layer skipping-only | 9.13 | 10.48 |
> | Learnable (ours) | 4.12| 6.28 |
>
>
> **[Q2: Extend to other Pre-trained Model Type]**
> Thank you for the suggestion. We validated our method on Minitron-4B [1], a representative efficient large language model of relatively small scale. Using the same hyperparameter settings as in our original experiments on Llama-3.1-8B, we trained the model for 10k iterations and evaluated the perplexity (PPL) on four resultant sub-networks. Table A2 below provides the results for this experiment.
>
> Table A2: Evaluation of our method using Minitron-4B as the pre-training model. We report the anchor model (the sub-models directly received gradient during training), as well as the interpolated model (the sub-models generated by router interpolation).
> | Model size | 25% (anchor) | 37.5% |  50% (anchor)  | 62.5% | 75% (anchor)  | 87.5% | 100% (anchor)  |
> |:-:|:-:|:-:|:-:|:-:|:-:|:-:|:-:|
> | PPL | 25.25 | 22.89 | 21.22 | 20.03 | 17.42 | 14.89 | 11.78 |
>
> [1] Compact language models via pruning and knowledge distillation

---

> ### Comment · Reviewer_7Wyq · 2024-11-26
>
> Thanks for your detailed response. I maintain my score.

---

> ### Author Response · Authors · 2024-11-29
>
> Many thanks for your thoughtful comments and suggestions. We value your support and will be incorporating all your feedback in our revised version.

---

### Comment · Area_Chair_gXjG · 2024-11-28
**Reviewers, please kindly respond**

Dear Reviewers,

If you have not responded to author's rebuttal, please kindly do so as soon as possible. The deadline is Dec 2, but the authors can potentially further clarify questions if you respond earlier. Thanks!

Best, AC

---

### Meta-Review · Area_Chair_gXjG · 2024-12-08

**Metareview:**

(a) Summary: introduces LLaMaFlex, a new weight-shared training architecture with routers, for LLMs. It enables zero-shot pruning for different widths and depths.

(b) Strengths: being able to "train once, deploy many"; strong performance; easy deployment with uniform architectures, etc.

(c) Weaknesses: limited evaluation (LLaMA 3.1 8B); clarity issues in writing/organization; more fair comparisons needed.

(d) Reasons for decision: reviewers' unanimous support; "train once, deploy many" is highly demanded for LLMs.

**Additional Comments On Reviewer Discussion:**

Authors added experiments on Minitron-4B, added ablations on importance metrics, clarified architectural constraints, reorganized sections, and included a limitations discussion. Reviewers acknowledged these, with one reviewer upgrading the rating.

---

### Decision · Program_Chairs · 2025-01-22

Accept (Poster)